# PRIVACY-PRESERVING MECHANISMS ENABLE CHEAP VERIFIABLE INFERENCE OF LLMS

## ABSTRACT

As large language models (LLMs) continue to grow in size, fewer users are able to host and run models locally. This has led to increased use of third-party hosting services. However, in this setting, there is a lack of guarantees on the computation performed by the inference provider. For example, a dishonest provider may replace an expensive large model with a cheaper-to-run weaker model and return the results from the weaker model to the user. Existing tools to verify inference typically rely on methods from cryptography such as zero-knowledge proofs (ZKPs), but these add significant computational overhead, and remain infeasible for use for large models. In this work, we develop a new insight – that given a method for performing *private* LLM inference, one can obtain forms of *verified* inference at marginal extra cost. Specifically, we propose two new protocols, each of which leverage privacy-preserving LLM inference in order to provide different guarantees over the inference that was carried out. Our approaches are cheap, requiring the addition of a few extra tokens of computation, and have little to no downstream impact. As the fastest privacy-preserving inference methods are typically faster than ZK methods, the proposed protocols also improve verification runtime. Our work provides novel insights into the connections between privacy and verifiability in LLM inference. We open-source our code at https://anonymous.4open.science/r/priveri/.

## 1 INTRODUCTION

Large language models (LLMs) have increased significantly in size over the last few years. Recent models achieving cutting-edge performance (DeepSeek-AI et al., 2025; Qwen et al., 2025; Team et al., 2025), for example, now often contain hundreds of billions of parameters. The hardware requirements to run these models are often too high for individuals, or even organizations, to run on their own, leading to a significant growth in demand for third-party LLM inference providers. However, this trend raises critical concerns about the integrity and trustworthiness of the services provided, particularly in the growing decentralized inference space. In this setting, any entity with surplus computational resources can offer to complete computational tasks, such as LLM inference, for another user. As the providers in this setting are often not subject to strict vetting, it is imperative to ensure that the service paid for is actually one that is performed by the provider.

Traditionally, the verification of outsourced computation has been addressed through cryptographic methods, such as zero-knowledge proofs (ZKPs). Although offering strong theoretical guarantees, these methods often introduce substantial computational overhead for either the prover (the inference provider) or the verifier (the user), or both. Despite significant progress in recent years, the state-of-the-art for ZK verification of LLM inference remains thousands of times slower than vanilla inference (Sun et al., 2024), rendering it infeasible for large models, which are particularly likely to be in demand for third-party inference provision.

A related concern for third-party compute provision is that of *privacy-preservation*. Performing LLM inference for another party requires the user to share their prompts, resulting in a loss of privacy. Therefore, a seemingly orthogonal line of work in recent years has focused on privacy-preserving computation. These include methods such as secure multi-party computation (SMPC) and fully homomorphic encryption (FHE).

Our work examines the question: **if a privacy mechanism is already in use, can this be leveraged to provide verification of the LLM inference computation as well?** We answer this question in

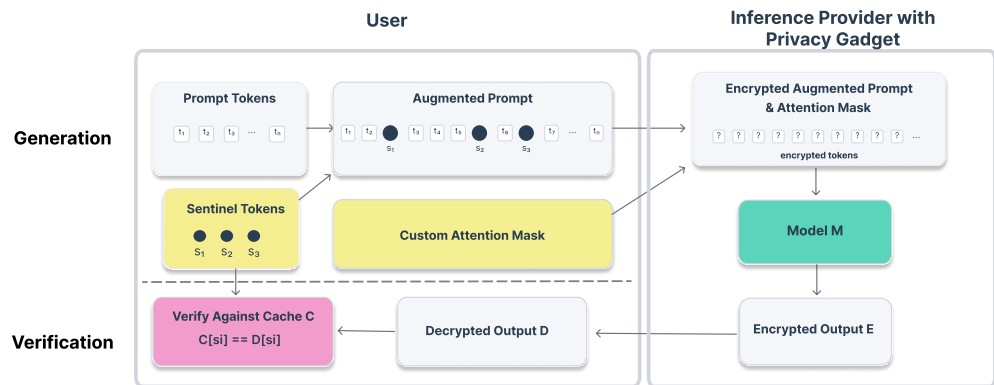

Figure 1: Schema of **Protocol 1**, which allows a user to verify that a specified model $M$ was run on their prompt by a third party inference provider. The user adds *sentinel tokens* in random positions to the original prompt tokens, along with an altered attention mask. The position of these sentinel tokens is obfuscated to the inference provider by the privacy-preserving mechanism. The user can therefore perform verification against a precomputed cache of the sentinel token logits, effectively ensuring that model substitution attacks cannot occur.

the affirmative; specifically, we propose two simple but novel protocols, 'logit fingerprinting', 'logit fingerprinting with noise', that use privacy to obtain differing levels of verification guarantees. We examine the costs and security properties of each of these protocols. Although our protocols have limitations and do not offer identical guarantees to those of ZK, we show that they are robust to many varieties of attacks. Moreover, we demonstrate that our 'logit fingerprinting with noise' protocol run with an SMPC method, SIGMA (Gupta et al., 2023), is $\sim 15\times$ faster than the state-of-the-art ZK method for proof of LLM inference on a single forward pass of Llama-2-7B (Touvron et al., 2023). We further hope that connecting privacy and verification for LLM inference will spur the creation of improved protocols and further research in this area; to this end, we also open-source an implementation of our Protocol 1 using the CrypTen library at https://anonymous.4open.science/r/priveri/.

## 2 BACKGROUND & RELATED WORK

**Privacy-preserving inference.** There are four main families of privacy-preserving LLM inference methods that have been proposed in the literature: SMPC (Secure Multi-Party Computation), FHE (Fully Homomorphic Encryption), TEEs (Trusted Execution Environments), and statistical methods. SMPC and FHE are general privacy-preserving computation methods which provide strong guarantees on computational indistinguishability of the inputs. Both methods add significant overhead to plaintext computation; for SMPC, a large component of this is communication between the multiple parties involved. Recently, both SMPC (Huang et al., 2022; Hao et al., 2022; Pang et al., 2023; Akimoto et al., 2023; Dong et al., 2023; Li et al., 2023) and FHE (Moon et al., 2024; Zhang et al., 2024) have been applied to LLM inference. Our protocols are agnostic to the exact method used, though differing attacks are possible with each choice of privacy mechanism. For a more detailed account of these methods, see Section A.1.

**Verifiable inference.** Zero-knowledge proofs (ZKPs) are a class of methods that allows one party, the prover, to prove to another party, the verifier, that a staement is true, without revealing any additional information beyond the proof itself. ZK methods have recently been applied to proving LLM inference, such as in Sun et al. (2024); Qu et al. (2025). However, these approaches have significant overhead, and remain thousands of times slower than vanilla inference. By contrast, recent work has introduced 'statistical' methods for verifiable LLM inference, where the guarantees are relaxed in order to reduce the overhead added. For a more detailed account of these methods, see Section A.2.

**Connections between privacy and verification.** The connection between privacy and verification has not been extensively studied previously. Perhaps the closest work is MPC-in-the-Head (Ishai et al., 2007), which introduced a zero-knowledge verification protocol by utilizing any SMPC protocol. The protocol comes with steep costs for both the prover and verifier. For example, the prover must not only locally simulate every party in the underlying MPC execution but also repeat the computation multiple times. On the verifier's end, the party must perform several confirmation tasks, including recomputing opened views, consistency checks, and typically engage in multiple rounds of checking to achieve acceptable soundness. The crucial distinction of our suggested protocols to MPC-in-the-Head is that we use the privacy scheme *directly* to encode inexpensive secrets that are easily verifiable. To the best of our knowledge, there has not previously been any work that specifically examines the relationship between privacy-preserving LLM inference and verifiable inference of LLMs in this way.

## 3  THREAT MODEL

We consider a setting with two primary roles: the **user**, who also acts as the verifier, and an **inference provider**, who also acts as the prover. The user wishes to run inference with a model $M$ on their prompts $x$, but cannot do so themselves due to e.g. lack of computational resources. They therefore request the inference provider to perform inference on $x$ with $M$. The inference provider is untrusted and may act as an adversary without behavioral constraints; other external adversaries are out of scope. We assume the use of a privacy-preserving mechanism providing computational indistinguishability of the inputs to ensure that the inference provider cannot view $x$. The model weights are assumed to be public. Our security goal is verifiability – that is, ensuring that the output the inference provider returns to the user can be verified as being the correct forward pass on the requested model on the given privatized prompt.

## 4  PROTOCOL 1: LOGIT FINGERPRINTING

Our first proposal for obtaining verification cheaply given access to a privacy-preserving method of LLM inference is **logit fingerprinting**. We hypothesize that the logit vector returned by performing a forward pass on any set of tokens on modern LLMs is a highly unique 'fingerprint' of the model. Our proposed protocol leverages this property to provide inference verification as follows (see Figure 1):

1. First, the user inserts $K$ *sentinel tokens* into the tokenized prompt, at random positions within the prompt. Call these positions $p_1, p_2, ..., p_K$. These $K$ tokens are taken randomly from a public cache $C$, consisting of many such length $K$ sequences.

2. Next, the user creates the 2D attention mask to be used by the LLM by taking their desired attention mask (e.g., lower triangular for decoder-only LLMs) and inserting rows and columns as follows.
   - Add a row at $p_i$ that is 0 everywhere except positions $p_j \, \forall \, j \leq i$, where it is set to 1.
   - Add a column at $p_i$ that is 0 everywhere except positions $p_j \, \forall \, j \geq i$, where it is set to 1.

3. The attention mask and augmented tokenized prompt are given to the inference provider under a privacy-preserving scheme, and the inference provider carries out a forward pass, and returns the output logit vector at all token positions to the user.

4. The user verifies that the sentinel token logits match against the precomputed cached logits for that specific model.

The construction of the attention mask is such that the sentinel tokens do not attend to, and are not attended by, any of the original prompt tokens, but they do attend to each other in standard autoregressive fashion. This also ensures that sentinel tokens have no downstream impact on the original prompt when inference is performed. A formal description of this procedure is given in Section D.

### 4.1  COST ANALYSIS

**Inference provider (prover).** Excluding the overhead of the private inference scheme, the total number of extra operations is a factor of $\frac{K}{N}$, where $N$ is the length of the original prompt. As we

discuss in Section 4.2, $K$ can be set to be as small as 3 and retain strong security properties, so this is very small for reasonably sized $N$. Furthermore, if the privacy scheme supports parallelized inference, this can result in almost no extra runtime.

**User (verifier).** The verifier is required to pick a sequence from a public cache and perform a matching on the returned logits against the same cache. The cost of this is minimal and does not require specialized hardware.

**Construction of the cache.** Constructing the cache entails an initial computational cost and also must be performed by a trusted party, since it underpins the correctness of the protocol. Ideally, this responsibility is delegated to an entity with sufficient computational resources to produce a verifiable proof of correctness, for example, in the form of a zero-knowledge proof. Although the computational expense of this might be significant, the cost is incurred only once and is then amortized across all subsequent inference calls, including potentially all prover-verifier pairs.

## 4.2 SECURITY ANALYSIS

In this section, we assume that logits are indeed unique fingerprints of models. We perform analysis across a range of models in Section 4.3 to verify this is the case.

In order for the inference provider to not be able to guess the logits to return for the sentinel tokens, the set of sentinel tokens must be randomly chosen from a large set of possibilities. The crux of this protocol is that the inference provider cannot determine which of the possibilities is specifically being asked for in any particular instance due to the privacy mechanism.

**Probabilistic attacks.** This protocol utilizes two elements of randomization: the choice of the sentinel tokens, and their positions. For the former, if the user selects the sequence uniformly at random from a cache of size $|C|$, then a dishonest inference provider can guess it with probability $1/|C|$. $|C|$ can therefore be set to desired tolerances. For the latter, under a privacy-preserving mechanism that also preserves tensor structure (such as SMPC), correctly guessing the sentinel tokens' exact positions is sufficient for a successful attack: the inference provider can perform the forward pass on only those components, and return arbitrary values for the other token positions. However, this occurs with probability $\binom{N+K}{K}^{-1}$, where $N$ is the length of the original prompt. When $K = 3$, for example, with $N = 14$, this is less than $1e-3$, and it drops further with increasing $N = 100$ to circa $1e-6$.

A related attack is to perform computation only on a random subset of the token indices. In the most extreme case, a dishonest provider takes $N + K - 1$ tokens, i.e. excludes exactly one token. The probability that all sentinel tokens are still selected (hence successfully passing verification) is $\frac{N}{N+K}$, requiring an infeasibly large $K$ to make secure – although it should be noted that in this case the dishonest provider is saving very little computation over honest behavior.

**Approximation attacks.** Another line of possible attacks are attempts by the inference provider to use a different model – especially, cheaper-to-run replacements – that still succeed in passing verification. Such alternatives could include smaller models from the same model family or approximations to the models by using e.g. low-rank projections of the weights. We perform experiments to test the robustness of the protocol to each of the above in Section 4.3 and find that verification fails immediately when any of the above are attempted.

## 4.3 EXPERIMENTS

**Setup.** We test the claim from Section 4.2 that pre–softmax logits can serve as model fingerprints. For each model $m$, we sample $N = 50{,}000$ token sequences of fixed length $K = 3$ from the model's token vocabulary (excluding special tokens). Given a sequence $\mathbf{t} = (t_1, t_2, t_3)$, we run a forward pass and record the next-token *logit vectors* at each position, $\ell_m^{(k)}(\mathbf{t}) \in \mathbb{R}^{V_m}$ for $k \in \{1, 2, 3\}$, where $V_m$ is the vocabulary size of model $m$. We define the *logit fingerprint*

$$\phi_m(\mathbf{t}) \;=\; \mathrm{concat}\big(\ell_m^{(1)}(\mathbf{t}),\, \ell_m^{(2)}(\mathbf{t}),\, \ell_m^{(3)}(\mathbf{t})\big) \in \mathbb{R}^{3V_m},$$

and compare fingerprints using L1 distance. We test on Llama 3.2 Instruct 1B, 3B, and 8B (Grattafiori et al., 2024), and on Qwen 2.5 Instruct 0.5B, 1.5B, 3B and 7B (Qwen et al., 2025). Comparisons are performed on FP32 logits; dropout is disabled.

| Model Approximation | Llama | Qwen |
|---|---|---|
| Same model, GPU non-determinism | 4.08 | 10.91 |
| Same model, different sequence | 2909 | 68326 |
| Cross-Model, same sequence | 329096 | 643719 |
| Low-rank factorization ($r = 2047$) | 833 | 7223 |
| Low-rank factorization ($r = 2040$) | 8029 | 124195 |
| Low-rank factorization ($r = 2000$) | 31194 | 216765 |
| 8-bit Quantization | 3499 | 8746 |
| Single step of finetuning | 471 | 1090 |

Table 1: L1 distances of logit fingerprints across different experimental settings for the Llama and Qwen families of models. **Honest behavior (GPU non-determinism) is clearly separated from dishonest behavior (all other rows).**

### 4.3.1 Honest Behaviors

**Floating point non-determinism.** We first provide context on the expected L1 distance due to non-determinism of floating-point operations (Shanmugavelu et al., 2024). This can be constituted as honest behavior; although, it is possible to also require an exact match, which would entail detecting hardware and batched-inference deviations. We run the *same* sequence multiple times with different batch sizes on GPU to measure this. We observe a maximum L1 deviation in doing so across all models tested of 10.90.

### 4.3.2 Dishonest Behaviors

We now test a wide range of dishonest behaviors and strategies.

**Intra-model.** Within each model, we compute the nearest-neighbor similarity among fingerprints from *distinct* sequences (i.e. $\mathbf{t} \neq \mathbf{s}$). Across $N = 50$k samples per model, there are no exact matches; the closest pair has an L1 distance of 2909.

**Within-family.** For the Llama family, the smallest L1 distance of logits we obtain is 329096. For the Qwen family, the minimum cross-model distance is 643719. These results indicate that even with a family of models, the logits are significantly different and suitable as fingerprints.

**Cross-family.** To enable comparisons across families with different vocabularies, we align dimensions by truncating the larger logit vectors to the smaller vocabulary size (i.e. comparing the first $\min(V_m, V_{m'})$ coordinates). Under this conservative alignment, Llama–Qwen comparisons exhibit substantially higher distances than the within-family maxima reported above (qualitatively, well above 800000).

**Low-rank factorization.** We approximate the linear layers of Llama 3.2 1B Instruct by replacing each weight matrix $W \in \mathbb{R}^{d_{\text{in}} \times d_{\text{out}}}$ with a rank-$r$ factorization $W \approx UV^\top$, where $U \in \mathbb{R}^{d_{\text{in}} \times r}$ and $V \in \mathbb{R}^{d_{\text{out}} \times r}$. The default hidden dimension of this model is 2048, so we test with $r \in \{2047, 2040, 2000\}$. Comparing fingerprints of 50k sequences between the full-rank and the low-rank variants, the minimum L1 distances observed are:

$$r = 2047 : 833.97, \quad r = 2040 : 8029.45, \quad r = 2000 : 31194.02.$$

Similarly, we obtained the following results for Qwen 2.5 3B Instruct:

$$r = 2047 : 7223, \quad r = 2040 : 124195, \quad r = 2000 : 216765.$$

**Quantization.** We next load Llama 3.2 1B Instruct and Qwen 2.5 3B Instruct in 8-bit precision using `bitsandbytes` and compare fingerprints to the full-precision (bfloat16) baseline. The minimum L1 distance is 3499 for Llama and 8746 for Qwen, again easily separated from the original model.

**Fine-tuning.** Finally, we evaluate robustness against model fine-tuning by comparing each of Llama 3.2 1B Instruct and Qwen 2.5 3B Instruct with a finetuned variant of each corresponding model on a single sample from FineWeb dataset for a single step. The minimum observed distance is 471 for Llama and 1090 for Qwen, consistent with the previous cases and again easily separable from the original model.

Our results are summarized in Table 1. The minimum L1 distance observed under dishonest behavior is 471, as seen in Llama's finetuning setting, while the maximum deviation with honest behavior due to floating point non-determinism is only 10.90 for Qwen. The significant difference allows for clear identification of honest vs. dishonest behavior; based on these results, we recommend using a matching threshold in the range of 15–20. Sequences whose logits differ by less than this threshold can be confidently regarded as originating from the same model; and even a single step of fine-tuning is easily detectable with this threshold.

## 4.4 LIMITATIONS

The main limitation of this protocol is that it can only be used to verify a single forward pass at a time, i.e. only generate a single new token, before requiring the user to repeat the protocol above with a fresh set of sentinel tokens and positions; otherwise, a dishonest provider could honestly perform the first forward pass (to pass verification) and provide spurious outputs for all subsequent forward passes. Thus, this protocol inherently requires user interaction for every step of token decoding. Another limitation is the vulnerability to the subsetting attack mentioned in Section 4.2. As such, we recommend that this protocol not be used in isolation with privacy mechanisms that retain tensor structure, such as SMPC methods.

## 5 PROTOCOL 2: LOGIT FINGERPRINTING WITH NOISE

The vulnerability of Protocol 1 to a subsetting attack reduces the space of privacy gadgets that it can be used with. Our second proposed protocol is designed to resist this attack. Our modification consists of adding randomly sampled noise to the token embeddings before they are passed into the LLM for the forward pass, and then using a lightweight predictor on the returned final hidden states to predict the noise that was used. Our proposed protocol is as follows:

1. First the user samples noise $b \in \mathbb{R}^d_e$, where $d_e$ is the embedding dimension of the model being used for inference, from a discrete set of possibilities $B$.

2. The user concatenates the noise to the embedding of the original prompt $e \in \mathbb{R}^{N \times d_e}$ in the $d_e$ dimension, to obtain a tensor $b_e \in \mathbb{R}^{N \times 2d_e}$.

3. The user applies the previously trained *NoiseEmbedder* module on $b_e$ to obtain $e' \in \mathbb{R}^{N \times d_e}$.

4. The user sends privatized $e'$, augmented with $K$ randomly-positioned sentinel tokens as in Protocol 1, to the inference provider.

5. The inference provider performs the forward pass and returns the final hidden states $h \in \mathbb{R}^{(N+K) \times d_h}$, where $d_h$ is the hidden dimension.

6. The user applies the logit-projection to the hidden states at the sentinel token positions, and checks the validity of these logits against the cache, as in Protocol 1.

7. The user applies the previously trained prediction module, the *NoisePredictor*, on $h$ at the non-sentinel positions to obtain estimated $\hat{b}$ at each such position. If each obtained $\hat{b}$ matches the sampled $b$ at that position, *and* the sentinel token logit check passes above, then the user can consider the inference to be verified.

In the above procedure, the sentinel tokens are not modified by the sampled noise, and so can be compared against the cache as in Protocol 1. For the remainder of the tokens, the predicted noise is compared to the sampled noise to verify that the forward pass was indeed carried out on each token position. The complete procedure is formally described in Section E.

## 5.1 COST ANALYSIS

**Inference provider (prover).** The cost to the inference provider is the same in this protocol as in Protocol 1.

**User (verifier).** In addition to the cost associated with the sentinel tokens, the user must now generate the sampled noise – which requires little computational cost – as well as run the NoiseEmbedder and NoisePredictor.

**Construction of the cache.** This cost remains the same as in Protocol 1.

**Training of NoiseEmbedder and NoisePredictor.** The NoiseEmbedder and NoisePredictor modules need to be trained for each different LLM in use. This entails an initial computational cost and also must be performed by a trusted party; however, similarly to the cache construction, this is a one-time cost that is then amortized over all subsequent inference calls on that model.

In Section 5.3, we show that the NoiseEmbedder and NoisePredictor can be simple linear projections, so that the additional cost to the user and the training cost can be made low in practice.

## 5.2 Security Analysis

We inherit the security analysis of Protocol 1 as it pertains to sentinel tokens – that is, sentinel tokens remain effective markers of the model that was used for the forward pass and are able to detect even very close replacements. For the non-sentinel tokens, the crux of the protocol's security now rests on the predictability of the injected noise.

Let the sample space size be given by $|B|$, and denote the accuracy of the prediction at token position $n$ under honest inference by $\mathrm{acc}_n := P(\hat{b}_n = b_n \mid \text{honest})$.

**Honest provider (completeness).** If the inference provider is honest, the probability that the user incorrectly rejects the returned computation is given by the probability that there is at least one mismatch in the predicted noise: $P(\text{incorrect rejection}) = 1 - \Pi_{n=1}^{N}\mathrm{acc}_n$.

**Dishonest provider (soundness).** If the inference provider is dishonest, the probability that the user incorrectly accepts the returned computation is given by the probability that $\hat{b}_n = b_n$ at all token positions $n$. Due to the privacy-preserving mechanism, the provider cannot know which $b_n$ was used, so the probability that $\hat{b}_n = b_n$ for any particular $n$ is upper bounded by $\frac{1}{|B|}$. In particular, the above implies that in a leave-one-out subsetting attack as described in Section 4.2, the probability of success is at most $\frac{1}{|B|}$.

## 5.3 Experiments

In this section, we describe a performant and lightweight architecture of the NoiseEmbedder and NoisePredictor; and we demonstrate their performance on Llama-3.2-1B, evaluating on the FineWeb-Edu dataset (Lozhkov et al., 2024).

**NoiseEmbedder architecture.** This module consists of a learned embedding $E \in \mathbb{R}^{|B| \times d_e}$, and a linear layer that is applied to the concatenation of the learned noise embeddings and the original embedding, and produces a single combined embedding as an output. Therefore the linear layer has a weight matrix: $W \in \mathbb{R}^{2d_e \times d_e}$.

**NoisePredictor architecture.** This module consists of a linear layer that takes the final hidden layer representations from the forward pass of the LLM and outputs unnormalized logits over the sample space $B$. Therefore the linear layer has a weight matrix: $W \in \mathbb{R}^{d_h \times |B|}$.

We fine-tune the NoiseEmbedder and NoisePredictor modules **whilst keeping the original model weights frozen**. As we are adding noise to the model embeddings, we train to optimize for both the log-likelihood on the dataset, as well as the classification accuracy of the NoisePredictor, using the cross-entropy loss. Our training objective is therefore given by:

$$\mathcal{L}_{\theta,\phi} \; = \; \mathbb{E}_{x,y\sim\mathcal{D}} \, \mathbb{E}_{b\sim B} \Big[ -\log f\big(y \mid \text{NoiseEmbedder}_\theta(x,b)\big) + \\ \lambda\, \text{CE}\big(\text{NoisePredictor}_\phi(f(\text{NoiseEmbedder}_\theta(x,b))),\, b\big) \Big] \tag{1}$$

where $x, y$ are the training data, f is the base model, $\theta$ and $\phi$ are the parameters of the NoiseEmbedder and NoisePredictor respectively, and $\lambda$ is a hyperparameter to be tuned. In practice, we find best results applying the same sampled noise to every token in the sequence. This does not impact the security analysis of Section 5.2. For further training and hyperparameter details, see Section F.

**Results.** Despite using a very lightweight NoiseEmbedder and NoisePredictor, and not modifying the original model weights at all, we find that we are able to achieve $\sim 99\%$ classification accuracy with $|B| = 100$ without *any* worsening of the log-loss on the given dataset. In particular, the base model's log-prob is $\sim 3.45$, and we achieve a held-out evaluation set log-prob of $\sim 3.43$ after training the modules, with noise injected.

### 5.4 LIMITATIONS

In comparison with Protocol 1, this protocol is resistant to subset attacks, due to the introduction of noise at each token position. However, this protocol adds extra computational burden to the user – they must now perform additional NoiseEmbedder and NoisePredictor forward passes. Although we have shown that these can be effective even if comprising just a single linear layer each, there may be some cases where even this extra computational requirement cannot be met. Moreover, the user must now also perform projection of the final hidden states to the logits themselves, necessitating another matrix multiplication. There is also now the additional computational requirement of training the modules prior to deployment, in a trust-secured manner. Finally, although we are able to achieve good accuracy rates of $99\%$ with $|B| = 100$, we have neither perfect soundness nor completeness; we hope that future work is capable of improving on the results we present here.

## 6 PROPERTIES UNDER DIFFERENT PRIVACY MECHANISMS

So far in the description of our protocols, we have remained largely agnostic to the privacy mechanism in use. In this section, we expand on the specific properties that our protocols possess under each of the main 3 privacy-preserving mechanisms: FHE, TEEs, and SMPC.

**FHE** The interaction of our protocols with FHE is the most straightforward. FHE has only a single party performing the encrypted inference. As shown by our analysis above, our protocols are capable of determining if that party is operating honestly or dishonestly.

**TEE** The interaction of our protocols with TEEs is multifaceted and complex. TEEs come equipped with both privacy-preservation (via memory encryption of processes restricted to hardware enclaves) as well as a form of verifiability with an attestation mechanism. Attacks on TEEs can degrade guarantees of the confidentiality, the attestation, or both. Our protocols can provide extra safety against attacks on attestation. In particular, the assumption of the trustworthiness of the hardware vendor in signing the enclave measurements can be relaxed. We provide further details in Section C.

**SMPC** SMPC protocols rely on multiple parties performing computations in order to obtain privacy-preserving inference. A typical assumption necessary for SMPC protocols to preserve their guarantees is that participants are 'honest-but-curious' i.e. they perform the prescribed computation faithfully, but they may use any means at their disposal in order to try to determine the content of the encrypted messages they receive. Our protocols allow for a relaxation of this (rather unrealistic) behavorial assumption. In particular, our protocols can detect the case where there are dishonest participants, as long as they do not all collude (if they do all collude, then privacy no longer holds in SMPC). Further analysis of this is given in Section B.

## 7 OUTPERFORMING STATE-OF-THE-ART ZK INFERENCE

We have shown that privacy-preserving mechanisms can enable verified inference. In this section, we describe the performance of privacy-preserving inference and our protocols, compared to the standard approach of zero-knowledge (ZK) proofs of inference.

The protocols we proposed in Section 4 and Section 5 are both compatible with FHE privacy schemes. However, state-of-the-art FHE schemes typically have greater overhead than ZK; for example, THOR (Moon et al., 2024) reports approximately 10 minutes for a single forward pass on an input of 128 tokens with BERT-Base (a model with 110M params), with GPU acceleration. By contrast, zkLLM reports just 74s of prover overhead for a forward pass with an input of 2048 tokens on OPT-125M. However, Protocol 2 (Section 5) is designed to resist tensor subset attacks, and is therefore also compatible for use with SMPC schemes. State-of-the-art SMPC schemes operate much faster than FHE.

We perform a direct comparison of our protocol with SMPC to zkLLM. We evaluate our protocol across varying generation lengths of {1, 2, 5, 10, 50, 100} tokens, on a 131 token input prompt, of which 128 are the original prompt tokens and 3 are sentinel tokens required by our construction. We run zkLLM on Llama-2-7B (Touvron et al., 2023) and measure the total prover time for these response lengths on a machine equipped with an A100 GPU. We compare these results to our own measurements of SIGMA (Gupta et al., 2023), a 2-party SMPC protocol optimized for GPU acceleration, which we evaluate on the same model and hardware. For zkLLM, we measure the runtime for a 128-token prompt.

Our protocol introduces a small, arguably negligible, communication overhead over SIGMA due to the need to transmit the encrypted augmented sequence, augmented attention mask, and augmented position identifiers to the inference provider. The total communication footprint is

$$b \cdot (L^2 + 8L + 15),$$

where $L$ is the sequence length and $b$ is the datatype size in bytes (e.g., 4 bytes for a `int32`). This expression accounts for the linear-size augmented prompt and position-id vectors and the quadratic-size augmented attention mask.

We evaluate under the same network assumptions as SIGMA. For the LAN setting, we assume a latency of $0.05\,\mathrm{ms}$ and a bandwidth of $9.4\,\mathrm{Gbps}$. For the WAN setting, we assume a latency of $60\,\mathrm{ms}$ and a bandwidth of $305\,\mathrm{Mbps}$. These settings ensure a consistent and fair comparison across network environments.

Our results for generation times are shown in Table 2. We see that Protocol 2 under SIGMA is approximately $\sim 15\times$ faster than zkLLM, across all response lengths; and that there is relatively little degradation in performance when moving from the LAN to the WAN network setting.

| Response Length | Ours with SIGMA (LAN / WAN) | zkLLM |
|:---:|:---:|:---:|
| 1 | 22.1 / 22.2 | 369.7 |
| 2 | 44.4 / 44.4 | 726.5 |
| 5 | 118.3 / 118.9 | 1798.1 |
| 10 | 227.7 / 236.6 | 3889.3 |
| 50 | 1194.0 / 1189.0 | 19550.8 |
| 100 | 2260.0 / 2301.0 | 35860.1 |

Table 2: Generation time (seconds) for Llama-2-7B with zkLLM vs. our Protocol 2 (SIGMA) for different response lengths under LAN and WAN conditions.

We also provide results for verification times in Table 3. Verification with our protocol consists of a lookup into the logit cache. This runs significantly faster than ZK verification.

| Response Length | Ours with SIGMA | zkLLM |
|:---:|:---:|:---:|
| 1 | 0.17 | 1.24 |
| 2 | 0.35 | 2.51 |
| 5 | 0.91 | 6.32 |
| 10 | 1.87 | 11.80 |
| 50 | 3.73 | 59.24 |
| 100 | 18.27 | 117.30 |

Table 3: Verification time (seconds) for Llama-2-7B with zkLLM vs. our Protocol 2 (SIGMA) for different response lengths.

**Discussion.** Our protocol as tested in a like-for-like setting is nearly $15\times$ faster than the state-of-the-art ZK method for proof of LLM inference. However, there are two key differences. First, ZK has fewer security assumptions. Although SMPC guarantees strong computational indistinguishability of its inputs in the non-colluding setting, it is vulnerable when all parties involved are dishonest and collude to pool their secret shares. By contrast, ZK is provably secure regardless of prover behavior assumptions. Second, our protocol still relies on statistical results, such as the accuracy of the NoisePredictor module. Therefore, our inference guarantees are not directly comparable to those produced by ZK methods.

Nevertheless, in settings where non-collusion can be ensured or encouraged, and where statistical guarantees are sufficient, our protocol offers a significant speedup over the state-of-the-art for proof of LLM inference.

## 8 CONCLUSION

We have introduced two protocols for verifying LLM inference, given the use of privacy-preserving mechanisms. These protocols are cheap for both the prover and the verifier and have little to no downstream impact. Future work may focus on mitigating the limitations of our protocols, for example by (1) boosting efficiency during many-token generation, (2) improving the statistical guarantees, or (3) guaranteeing resistance to attacks. We believe that connecting privacy and verifiability, particularly in LLM inference, will inspire future work on new and improved protocols.

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

# A BACKGROUND AND RELATED WORK

In this section, we provide a brief background on general methods of privacy-preserving function computation, general methods of verification, and their application to LLM inference in particular.

## A.1 PRIVACY-PRESERVATION

There are four main families of privacy-preserving inference of LLMs that have been proposed in the literature: **SMPC** (Secure Multi-Party Computation), **FHE** (Fully Homomorphic Encryption), **TEEs** (Trusted Execution Environments), and **statistical methods**. Here we provide brief background on each of these.

**SMPC** SMPC protocols split the required computation among multiple parties. The key ideas were originally developed in the 1980s (Yao, 1982; Goldreich et al., 1987) and provide mathematical guarantees that no single party can reconstruct the data on their own. Recently, the methodologies of SMPC have been applied to LLMs (Huang et al., 2022; Hao et al., 2022; Pang et al., 2023; Akimoto et al., 2023; Dong et al., 2023; Li et al., 2024). A difficulty uniformly faced by these protocols is efficient computation of the many non-linearities present in transformer-based LLMs; most of the works attempt to ameliorate this by using piecewise polynomial approximations which are more well-suited for MPC algorithms. However, this approximation leads to degraded inference results, and remains more expensive than direct computation of the non-linearities. The requirement of multiple parties also engenders significant communication overheads, and the further non-collusion requirement among the parties may be difficult to guarantee.

**FHE** FHE protocols require only a single party and make use of cryptographic methods to ensure that the result of the computation on the ciphertext is the same as that performed on the plaintext. The adjective 'fully' indicates the capability of performing arbitrary computations, not limited to a particular type or complexity. The first plausible construction of an FHE scheme was described in Gentry (2009); a more modern and widely used incarnation is CKKS (Cheon et al., 2017). Recently, CKKS has been further optimized and applied to LLM inference (Moon et al., 2024; Zhang et al., 2024), but similar issues arise with the non-linearities as SMPC methods. The overheads both for linear and non-linear operations are typically even larger than those in the SMPC setting.

**TEEs** Trusted Execution Environments (TEEs) (Sabt et al., 2015; Narra et al., 2019) create secure and isolated enclaves at the hardware level. This ensures confidentiality via memory encryption – allowing only the process running in the enclave to read the data. Furthermore, TEEs support integrity via attestation mechanisms. However, a significant concern is the vulnerability to side-channel attacks (Jauernig et al., 2020). Furthermore, attestation is only provided at boot-time and is not equivalent to an ongoing verification process. This process typically involves the TEE measuring the code and its environment, signing these measurements cryptographically, and sending a report for external verification. However, this is often a one-time check at the start and does not guarantee the integrity of the TEE throughout its execution. Finally, in cloud environments, attestation can rely on the cloud provider's services, which means users must trust the provider's proprietary attestation process without full transparency. This introduces a level of trust in the cloud provider's integrity, as these attestation services can be opaque "black boxes" that are not open to external audit. Moreover, there may be no independent way to verify the boot measurements provided by the cloud provider's infrastructure. The interaction between our protocol and TEEs is further discussed in Section C.

**Statistical Methods** A more broad and diverse grouping than the above is what we term 'statistical methods'. These are protocols without the mathematical guarantees of FHE or SMPC approaches, or the hardware-based guarantees of TEEs, but that instead employ statistical or empirical arguments to support the difficult of reversing ciphertext. Some ideas in this domain include the use of permutation-based security (Zheng et al., 2024; Yuan et al., 2024; Luo et al., 2024) or token-sharding based security (Thomas et al., 2025). These methods typically trade off the stronger guarantees of the above methods for greatly reduced overheads, sometimes approaching similar speeds to vanilla inference.

## A.2 VERIFICATION

**Zero-Knowledge Proofs (ZKP)** ZKPs are a class of methods that allows one party (the prover) to prove to another party (the verifier) that a statement is true, without revealing any additional

information beyond the proof itself. The main properties that ZKPs satisfy are completeness (an honest prover can convince a verifier that they performed the work as stated), soundness (a dishonest prover cannot convince a verifier that they performed the work), and the zero-knowledge property of not revealing any further information than the fact the work was done as stated. The first ZK protocol was introduced in 1985 in Goldwasser et al. (1985). Recently, ZK methods have been applied as proofs of inference for machine learning models, and specifically LLMs, in works such as Sun et al. (2024); Qu et al. (2025). However, these approaches remain thousands of times slower than vanilla inference – for example, zkLLM takes 15 minutes for generating a proof of a single forward pass for Llama-2-13B, compared to milliseconds for vanilla inference.

**Statistical Methods** Analogously to statistical methods of privacy-preservation, very recent work has investigated methods of relaxing the standard of proof of work provided in order to reduce computational overhead. Ong et al. (2025) encodes and validates the most salient features of the last hidden state tensor of an LLM using a compact, verifiable proof, which is then recomputed in parallel by the verifier. Although the authors demonstrate how to set up a commitment scheme that has relatively little overhead to the prover, and verification is faster than full recomputation thanks to parallelization, there is still a requirement for the verifier to perform a full LLM forward pass, potentially necessitating specialized hardware. Sun et al. (2025) proposes the use of a 'proxy task' based on the last hidden layer features of an LLM that can then be utilized by the user to compare to a label that they would expect based on their original input. The method proposed requires trust assumptions from the platform for generation of the proxy-task feature extractor and labeller networks, as well as secret generation/embedding, and adds the overhead of computation to perform all of the above.

## B  INTERACTION OF PROTOCOLS WITH SECURE MULTIPARTY COMPUTATION (SMPC)

In Section 6, we discussed how our protocols can detect dishonest behavior in the case where not all parties collude, under an SMPC privacy-preserving scheme. Here, we break down the various behavioral cases in detail:

**Case 1:** If there is at least a single honest party participating, that party will be able to detect any dishonest party attempting to deviate from the protocol by trying to evaluate secret shares of the private prompt on an incorrect circuit (i.e., wrong model). However, that honest party may not be incentivized to report this dishonesty; our protocols ensure that this dishonest behavior is detected, even if the honest party does not report the deviation.

**Case 2:** If all parties are dishonest and colluding, then the privacy-preservation property of SMPC is broken, and our protocols are no longer effective.

**Case 3:** If all parties are dishonest but at least one party is non-colluding, then for most SMPC protocols, privacy-preservation holds. In this setting, our protocols can still detect the dishonest behavior.

In summary, in Case 1, our protocols ensures detection of dishonest behavior regardless of incentive structures; and in Case 3, SMPC by itself is insufficient to ensure the correctness of the computation, and our protocols provide this surety, thereby extending the range of behaviors under which faithful computation can be ensured with SMPC.

## C  INTERACTION OF PROTOCOLS WITH TRUSTED EXECUTION ENVIRONMENTS (TEEs)

Trusted Execution Environments (TEEs) provide confidentiality through encrypted and isolated execution, and offer a hardware-backed attestation mechanism intended to certify the integrity of the running code. Because our protocol is designed to compose with general privacy-preserving execution mechanisms, TEEs naturally fit within the class of privacy gadgets through which our approach can achieve verifiability.

However, TEE attestation has well-documented limitations. As discussed in Section A.1, numerous attacks have been demonstrated across multiple TEE architectures, and additional vulnerabilities

continue to emerge. These attacks have different properties w.r.t. whether they break confidentiality, verifiability, or both, and the exact extent to which they do either (e.g. some attacks allow for a limited degree of memory modification, but not sufficiently so to replace the entirety of a model's loaded weights in a VM). Given the wide variety of attacks, and the very particular details around the adversarial behaviors they enable, we refrain in this paper from making broad statements regarding how our protocols would interact with TEE attack surfaces.

A notable structural assumption of TEE-based attestation is that the hardware vendor acts as the root of trust; the system relies on the vendor not signing forged or incorrect enclave measurements. Our protocol reduces reliance on this assumption: by providing an independent cryptographic verification layer, we relax the trust placed in attestational integrity and enable a lower-trust deployment model in which TEEs serve primarily as a confidentiality mechanism rather than as the sole source of correctness.

Finally, we note that TEEs remain appealing in practice because their runtime overhead is substantially lower than that of fully cryptographic approaches such as SMPC or FHE (typically only 5–10% for LLM inference). In combination with our protocol, TEEs can therefore support a hybrid design that combines their lightweight confidentiality guarantees with cryptographic verifiability that does not depend exclusively on vendor-rooted trust.

## D    FORMAL ALGORITHM FOR PROTOCOL 1

The procedure is comprised of three components: cache generation through Algorithm 1, inference request through Algorithm 2, and the verification stage through Algorithm 3.

---

**Algorithm 1** Cache Generation

---

**Input:** model, cache size $|C| \in \mathbb{N}$, sentinel token count $K \in \mathbb{N}$
**Output:** cache: mapping $s \mapsto \ell_{1:K} \in \mathbb{R}^{K \times V}$
 1: cache $\leftarrow \emptyset$
 2: **while** $|cache| < |C|$ **do**
 3:     $s_{1:K} \leftarrow$ sample with replacement $K$ tokens from $V$
 4:     $m_{1:K,1:K} \leftarrow 0$                        $\triangleright$ initialize $K \times K$ attention mask
 5:     **for** $i = 1..K$ **do**
 6:         **for** $j = 1..i$ **do**
 7:             $m_{i,j} \leftarrow 1$
 8:         **end for**
 9:     **end for**
10:     $\ell \leftarrow$ model.forward$(s, m)$                        $\triangleright \ell \in \mathbb{R}^{K \times V}$
11:     cache$[s] \leftarrow \ell$
12: **end while**
13: **return** cache

---

---

**Algorithm 2** Inference Request

---

**Input:** prompt token embeddings $x_{1:N}$, attention mask $a_{1:N}$, sentinel token sequence $s_{1:K}$

**Output:** logits $\ell_{1:N+K} \in \mathbb{R}^{(N+K) \times V}$

1: positions $p_{1:K} \leftarrow$ sample without replacement $K$ times from Uniform$[1, N+K]$
2: augmented embeddings $x'_{1:N+K} \leftarrow$ insert sentinel tokens $s_{1:K}$ at positions $p_{1:K}$
3: augmented mask $a'_{1:N+K} \leftarrow$ expand $a_{1:N}$ at positions $p_{1:K}$ with 0-filled rows and columns
4: **for** i = 1..K **do**
5:     **for** j = 1..i **do**
6:         $a'[p_i, p_j] \leftarrow 1$
7:     **end for**
8:     **for** j=i..K **do**
9:         $a'[p_j, p_i] \leftarrow 1$
10:     **end for**
11: **end for**
12: $x' \leftarrow$ encrypt($x'$)
13: $a' \leftarrow$ encrypt($a'$)
14: encrypted $\ell_{1:N+K} \leftarrow$ inference provider forward pass on encrypted $x', a'$
15: **return** decrypt($\ell_{1:N+K}$)

---

**Algorithm 3** Verification

---

**Input:** logits $\ell_{1:N+K} \in \mathbb{R}^{(N+K) \times V}$, sentinel positions $p_{1:K} \subset \{1, 2, ..N+K\}$, sentinel sequence $s_{1:K}$, cache $\in \mathbb{R}^{C \times K \times V}$, tolerance $tol > 0$

**Output:** verified: bool

1: verified $\leftarrow$ true
2: **for** $i = 1..K$ **do**
3:     $p' \leftarrow p[i]$
4:     $err \leftarrow \|\ell[p'] - \text{cache}[s][i]\|_1$
5:     **if** $err > tol$ **then**
6:         verified $\leftarrow$ false
7:     **end if**
8: **end for**
9: **return** verified

---

# E  FORMAL ALGORITHM FOR PROTOCOL 2

The procedure is also comprised of three algorithms similar to those in Section D: noised embedding generation Algorithm 4, noisy inference request Algorithm 5, and verification with noisy prediction Algorithm 6.

---

**Algorithm 4** Noised Embedding Generation

---

**Input:** discrete noise set $B$, trained NoiseEmbedder $E$, embedding dim $d_e$, token embedding $e \in R^{d_e}$
**Output:** sampled noise $b \in R^{d_e}$, noised embedding $e_b \in \mathbb{R}^{d_e}$
1: $b \leftarrow$ sample one value uniformly from $B$
2: $b_e \leftarrow$ concat$(e, b)$                                                           ▷ $b_e \in \mathbb{R}^{2d_e}$
3: $e' \leftarrow E$.forward$(b_e)$                                                          ▷ $e' \in \mathbb{R}^{d_e}$
4: **return** $(b, e')$

---

**Algorithm 5** Noisy Inference Request

---

**Input:** prompt token embeddings $x_{1:N}$, attention mask $a_{1:N}$, sentinel token sequence $s_{1:K}$
**Output:** hidden states $h \in \mathbb{R}^{(N+K) \times d_h}$, noise_cache $\in B^N$
1: positions $p_{1:K} \leftarrow$ sample without replacement $K$ times from Uniform$[1, N + K]$
2: augmented embeddings $x'_{1:N+K} \leftarrow$ insert sentinel tokens $s_{1:K}$ at positions $p_{1:K}$
3: augmented mask $a'_{1:N+K} \leftarrow$ expand $a_{1:N}$ at positions $p_{1:K}$ with 0-filled rows and columns
4: **for** i = 1..K **do**
5:     **for** j = 1..i **do**
6:         $a'[p_i, p_j] \leftarrow 1$
7:     **end for**
8:     **for** j=i..K **do**
9:         $a'[p_j, p_i] \leftarrow 1$
10:    **end for**
11: **end for**
12: **for** each non-sentinel token $t$ in $x'$ **do**
13:     $b, e' \leftarrow$ call Algorithm 4 on $x'[t]$
14:     $x'[t] \leftarrow e'$
15:     noise_cache[t] $\leftarrow$ b
16: **end for**
17: $x' \leftarrow$ encrypt$(x')$
18: $a' \leftarrow$ encrypt$(a')$
19: encrypted $h_{1:N+K} \leftarrow$ inference provider forward pass on encrypted $x', a'$
20: $h \leftarrow decrypt(h_{1:N+K})$
21: **return** $h$, noise_cache

---

---

**Algorithm 6** Verification With Noise Prediction

---

**Input:** decrypted hidden states $h_{1:N+K} \in \mathbb{R}^{(N+K) \times d_h}$, sentinel positions $p_{1:K}$, sentinel sequence $s_{1:K}$, logit_cache $\in R^{C \times K \times V}$, noise_cache $\in B^N$, NoisePredictor $NP$, logit projection $L : \mathbb{R}^{d_h} \to \mathbb{R}^V$, sentinel tolerance $tol_s$

**Output:** verified: bool

1: **for** $i = 1..K$ **do**
2:     $p' \leftarrow p_i$
3:     $\ell \leftarrow L(h[p'])$
4:     $err \leftarrow \|\ell - \text{logit\_cache}[s][i]\|_1$
5:     **if** $err > tol_s$ **then**
6:         verified $\leftarrow$ false
7:     **end if**
8: **end for**
9: **for** $j = 1..N + K$ **do**
10:     **if** $j \in P$ **then**
11:         **continue**
12:     **end if**
13:     $\hat{b} \leftarrow NP.\text{forward}(h[j])$
14:     **if** $\hat{b} \neq \text{noise\_cache}[j]$ **then**
15:         verified $\leftarrow$ false
16:     **end if**
17: **end for**
18: **return** verified

---

## F   TRAINING DETAILS FOR PROTOCOL 2

In this section we provide further details for the experiments conducted in Section 5.3.

We train on the FineWeb-Edu dataset (Lozhkov et al., 2024). This is a large-scale dataset of 1.3T total tokens consisting of high-quality educational web pages filtered from the larger FineWeb dataset. This dataset has been used for pretraining, and is suitable for general testing of language modeling capabilities of LLMs. We take 40000 samples from this dataset and divide these into an 80/20 train-validation split. We perform training for 500 steps with a batch size of 64 on sequences of length 256. We use the optimizer AdamW (Loshchilov & Hutter, 2019) with a learning rate of $5e - 4$, with no warmup steps. The base model is Llama-3.2-1B, and the weights of this model are frozen; gradients are backpropagated through this model in order to reach the NoiseEmbedder module.

We utilize the same sampled noise at every token position, but a different noise is sampled for each batch element. Our results are reported using a $\lambda$ hyperparameter value of 3.5. We train using an A100 GPU.

