# OpenReview forum: "Privacy-Preserving Mechanisms Enable Cheap Verifiable Inference of LLMs"
_ICLR.cc/2026/Conference — Submitted to ICLR 2026_

### Official Review · Reviewer_NATw · 2025-10-24

**Soundness:** 2
**Presentation:** 3
**Contribution:** 2
**Rating:** 2
**Confidence:** 3

**Summary:**

The paper focus on the problem of verifiability of hosted LLM inference services. Different from prior works that use heavy ZKP technologies, the authors propose to verify the LLM output by inserting random tokens to the user prompt or adding random noises to the token embeddings, where the output logits can be significantly different if a different (typically smaller) model is used. The user-side perturbation is made oblivious to the server owing to the assumed underlying MPC-based inference. The proposed methods exhibit both theoretical and statistical resilience to attacks.

**Strengths:**

- The paper is well-written and easy to follow.
- The problem of verifying MLaaS is an increasingly important direction.
- The idea of build verification upon MPC-based secure inference is interesting.

**Weaknesses:**

- Table 2 only shows experiments for one single forward pass. However, the proposed approaches  require continual user interaction at every decoding step. Runtime for a full generation is need to showcase the introduced additional cost.
- Protocol 1 and Protocol 2 both inherently require user interaction for every step of token decoding. This is not reasonable in real-world deployment,
- Protocol 3, which is claimed to be non-interactive, also has a significant limitation. For any models with instruction-following, the key appending does not work.

**Questions:**

Please refer to the weakness. I belive the current limitations for each protocol is significant and not applicable to real-world deployment.

---

> ### Author Response · Authors · 2025-11-21
>
> Thank you very much for your detailed review. We are glad that you found our paper has good presentation, that our idea is interesting, and that our paper tackles an important problem.
>
> We address your other points below.
>
> W1 & W2: Thank you for raising this key point! We agree that it is important to test for longer generations than a single token. We have now performed further experiments to do so. We have extended the experiment in Section 7 to include generation of 1, 2, 5, 10, 50 and 100 tokens, and compared the total time taken overall for our protocol to zkLLM. We further break down the total time taken for the prover, the verification, and also include an ablation across different network speed settings.
>
> We use the same network conditions as SIGMA, namely a latency of 0.05ms and a bandwidth of 9.4 Gbps for LAN connections, and a latency of 60ms and a bandwidth of 305 Mbps for WAN connections. See the table below for the full results for a prompt of 128 tokens on Llama 2 7b.
>
>
> | **Response Length** | **Ours + SIGMA – Generation (LAN / WAN)** | **zkLLM – Generation** |
> |---------------------|--------------------------------------------|--------------------------|
> | 1   | 22.1 / 22.2       | 369.7   |
> | 2   | 44.4 / 44.6       | 726.5   |
> | 5   | 118.3 / 118.9     | 1798.1    |
> | 10  | 227.7 / 236.6     | 3889.3    |
> | 50  | 1189.0 / 1194.0   | 19550.8   |
> | 100 | 2260.0 / 2301.0   | 35860.1   |
>
> | **Response Length** | **Ours + SIGMA – Verify** | **zkLLM – Verify** |
> |---------------------|---------------------------|---------------------|
> | 1   | 0.17   | 1.24   |
> | 2   | 0.35   | 2.51   |
> | 5   | 0.91   | 6.32   |
> | 10  | 1.87   | 11.8   |
> | 50  | 3.73   | 59.2   |
> | 100 | 18.27  | 117.3  |
>
> As can be seen, our approach remains significantly faster for both generation and verification, even in the slower WAN setting, for long output responses, than the state of the art comparison, zkLLM.
>
>
> W3: Our Protocol 3 was tested extensively with instruction-following models, and so our experiments explicitly show that it does work with such models, so we do not quite understand your statement. Could you please clarify this for us?
>
> Finally, we would like to draw to your attention that we have now open sourced an implementation of our Protocol 1 using the CrypTen library at https://anonymous.4open.science/r/priveri/.
>
> Thank you once again for your review and your insightful comments. We have made a significant effort to extend our experimental section to address your main concerns, and would appreciate it if you would consider raising your score in light of our response. We are also happy to continue the discussion on any other questions or points you have, until the end of the discussion period. Thank you!

---

> > ### Comment · Reviewer_NATw · 2025-11-27
> >
> > Thank the authors for the detailed response. Several of my concerns have been addressed. However, I still have a few questions:
> >
> > - Could the authors provide the communication size between the verifier and prover? IMO, sending back logits rather than decoded text may cause huge communication overhead.
> > - Regarding W3, sorry for the typo. I am actually wondering the effectiveness of Protocol 3 on models without strong instruction-following capabilities. As shown in Table 3, smaller models tend to exhibit lower transcription success rates.

---

> > > ### Author Response · Authors · 2025-11-27
> > >
> > > Thank you for the continued engagement with our work. We appreciate your follow-up questions and are pleased to clarify the remaining points.
> > >
> > > **Q1: Communication Cost**
> > >
> > > We measured the communication footprint between the verifier and prover by accounting for all tensors transmitted between them: the augmented input sequence, the attention mask, the position ids, and the logits. We used the identical model, prompt lengths, response lengths, and network settings reported previously (Llama-2-7B, 128-token input, vocabulary size 32k). The full results appear below:
> > >
> > > | N (tokens) | Comm Size (GB) | LAN Time (s) | WAN Time (s) |
> > > | ---------- | -------------- | ------------ | ------------ |
> > > | 1          | 0.03           | 0.00         | 0.17         |
> > > | 2          | 0.06           | 0.01         | 0.34         |
> > > | 5          | 0.15           | 0.02         | 0.84         |
> > > | 10         | 0.31           | 0.04         | 1.68         |
> > > | 50         | 1.54           | 0.18         | 8.42         |
> > > | 100        | 3.08           | 0.36         | 16.83        |
> > >
> > > While modern models indeed tend to adopt larger vocabularies, the associated increase in communication remains modest relative to the overall inference cost. In all practical settings we evaluated, communication between the verifier and prover accounted for only a small fraction of total runtime (< 1%) and therefore does not constitute a bottleneck for the protocol.
> > >
> > > **Q2: Protocol 3 for Non-Instruct Models**
> > >
> > > Yes, we agree that this is a limitation of Protocol 3. In particular, the protocol is unlikely to work with base non-instruct-tuned models. However, instruction-tuned models are by far the most commonly used in hosted-inference settings, so we do not view this as a significant practical limitation. Furthermore, as we show in Table 3 in Appendix E, all models that we tested that were above 3B parameters in size were capable of performing the task very well (>98% of the time), so we find that this requirement is satisfied by many models in practice.
> > >
> > > However, based on the feedback of other reviewers, we have now removed Protocol 3 from our paper. We kindly request that you consider our paper’s contributions based on Protocol 1 and 2, which form the bulk of our submission and experimental work.
> > >
> > > Once again, thank you for your constructive and thoughtful feedback throughout the discussion period. We hope the clarifications above address your remaining concerns, and we would be grateful if you might consider adjusting your evaluation accordingly.

---

### Official Review · Reviewer_eGgA · 2025-10-30

**Soundness:** 2
**Presentation:** 1
**Contribution:** 2
**Rating:** 4
**Confidence:** 1

**Summary:**

This paper addresses a critical practical challenge in remote large language model (LLM) inference: untrusted third-party providers may substitute expensive, high-capacity models with cheaper, weaker alternatives (e.g., replacing LLaMA-70B with LLaMA-7B) to cut costs, while users lack mechanisms to verify the authenticity of the computation. Existing verification tools (e.g., zero-knowledge proofs, ZKPs) are infeasible for LLMs due to prohibitive computational overhead.  The authors propose three distinct protocols that leverage privacy-preserving inference to provide targeted verification guarantees. The work aims to advance both practical verification tools for remote LLM inference and theoretical understanding of privacy-verifiability links in AI systems.

**Strengths:**

1. This paper addresses a high-priority problem in LLM deployment with a novel, practical insight: leveraging private inference to enable low-overhead verification.
2. The link between private and verified inference is a key innovation.

**Weaknesses:**

1. The presentation of this paper is not good enough, and it is a little hard to understand how it works. Maybe a schematic graph helps
2. What are connections between three protocols? How does this paper relate to existing works?

**Questions:**

see weaknesses

---

> ### Author Response · Authors · 2025-11-21
>
> Thank you very much for your review. We are glad that you found our paper addresses an important problem and that our proposed solution is innovative.
>
> We address your other points below:
>
> W1: Thank you for raising this point. We have now included a figure corresponding to our Protocol 1, and forming the basis of Protocol 2, which can be found [here](https://ibb.co/7dxKHbc5). We have also now included these figures in our updated draft. We hope that these can help clarify our protocols.
>
> W2: Thank you for mentioning this point. Protocol 2 is an extension of Protocol 1, used to address a potential vulnerability in the SMPC setting, as we mention in line 266. Protocol 3 is a separate idea entirely.
>
> In terms of previous work, as we mention in Section 2, there is almost no prior work relating privacy to verifiability that we are aware of. The closest such idea is MPC-In-The-Head, which utilizes any SMPC protocol to obtain a corresponding zero-knowledge protocol. However, our work differs in that MPC-In-The-Head does not directly encode material using the privacy method to assist in the proving process. Furthermore, as far as we are aware, we are the first to suggest protocols that are suitable for LLM inference specifically, and test performance on them.
>
> Finally, we would like to draw to your attention that we have now open sourced an implementation of our Protocol 1 using the CrypTen library at https://anonymous.4open.science/r/priveri/. Further, we have extended our experiments in Section 7 to longer response lengths and different network conditions -- please see our response to Reviewer Y7kc for the details. We find that our protocols remain significantly faster for both generation and verification than zkLLM.
>
> Thank you once again for your thorough review and your insightful comments. If you have any further questions or comments for any part of our responses, or if you think our submission can be further improved in any way, please let us know. We are happy to continue the discussion at any time until the end of the discussion period. We have made a significant effort to address each of your questions and weaknesses and would appreciate it if you would consider raising your score in light of our response. Thank you!

---

### Official Review · Reviewer_Y7kc · 2025-11-01

**Soundness:** 3
**Presentation:** 2
**Contribution:** 3
**Rating:** 6
**Confidence:** 2

**Summary:**

This paper explores the intersection between privacy-preserving computation and verifiable inference for large language models (LLMs). It argues that when privacy-preserving techniques such as Secure Multi-Party Computation (SMPC) or Fully Homomorphic Encryption (FHE) are already used for model inference, these mechanisms can be extended to provide verifiable computation almost for free.

To support this idea, the paper proposes three novel protocols: Logit Fingerprinting, Logit Fingerprinting with Noise, and Key Appending. Experimental results show that these methods can detect dishonest inference efficiently and run much faster than zero-knowledge proof (ZK)–based approaches.

Since I am not an expert in security or cryptographic verification, I may not be able to fully assess the depth or novelty of the proposed mechanisms. However, from a general machine learning perspective, the idea of connecting privacy and verification in LLM inference seems potentially impactful.

**Strengths:**

1. This work presents a creative and timely exploration of a relatively unstudied relationship between privacy and verifiability in large model inference.

2. The motivation is clear, as the growing reliance on third-party model hosting introduces both privacy and integrity risks.

3. The three proposed protocols provide a spectrum of practical trade-offs between interaction cost, computational efficiency, and verification strength.

**Weaknesses:**

1. The paper seems to rely mostly on empirical results rather than formal proofs, and it is not clear how strong the guarantees are compared with existing cryptographic approaches.

2. The evaluation, while comprehensive in experiments, could include more discussion of practical deployment aspects such as latency, communication overhead, and integration into existing systems. It is also not entirely clear how well the proposed methods scale to long or interactive LLM sessions.

3. For readers without a strong security background, it would be helpful to have clearer explanations of key assumptions (for example, what it means for a provider to be “honest but curious,” or how the privacy mechanism ensures non-collusion).

**Questions:**

- Could the authors clarify, in more intuitive terms, what kind of verification guarantee each protocol provides compared with standard cryptographic verification? For instance, are these probabilistic or statistical guarantees?
- How do the proposed methods perform in more realistic, multi-turn generation settings where verification cannot be repeated at every step?
- To what extent can these protocols be combined with lighter-weight security assumptions such as trusted execution environments (TEEs)?
- From an engineering perspective, how difficult would it be to integrate these protocols into existing inference frameworks used in industry?
- Could the authors discuss more concretely what kind of real-world scenarios or users would most benefit from these methods?

---

> ### Author Response · Authors · 2025-11-21
> **Response Part 1 to Reviewer Y7kc**
>
> Thank you very much for your detailed review. We are glad that you found our paper creatively addresses a well-motivated and timely problem, and that our proposed protocols cover a diversity of benefits and tradeoffs.
>
> We address your other points below.
>
> W1 & Q1: Thank you for mentioning this point. Yes, our work provides statistical, rather than cryptographic, guarantees, which are necessarily weaker. However, we would point out an analogy to the adversarial robustness literature; in that field, the most performant algorithms (e.g. see [RobustBench](https://robustbench.github.io/)) generally do not have provable worst case guarantees, but are nevertheless the most useful and resistant to attacks in practice.
>
> W2 & Q2: We wish to clarify that Protocol 1 and 2 both require user interactivity at every step. Therefore, if there are T new tokens generated, there need to be T verifications conducted. These verifications may be performed one at a time, before every new token is generated; or they may be performed asynchronously, while the next token is being generated; or they may all be collated into a batch verification step that is performed together at the end of generation. It is possible to relax this requirement – for example, by only applying sentinel tokens during random generation requests, and utilizing economic incentives to dissuade a dishonest party from taking chances.  For example, if there is a significant penalty applied to the inference provider for a failed verification, then a rational agent will not attempt dishonest behavior even if they are able to succeed in doing so undetected with some probability, as long as the penalty is high enough. However, this entails a different set of considerations from our setting, so we do not further explore this idea.
>
> We agree that it is important to clearly indicate the communication and latency overhead of the above. We have now performed further experiments to do so. We have extended the experiment in Section 7 to include generation of 1, 2, 5, 10, 50 and 100 tokens, and compared the total time taken overall for our protocol to zkLLM. We further break down the total time taken for the prover, the verification, and also include an ablation across different network speed settings.
>
> We use the same network conditions as SIGMA, namely a latency of 0.05ms and a bandwidth of 9.4 Gbps for LAN connections, and a latency of 60ms and a bandwidth of 305 Mbps for WAN connections. See the table below for the full results for a prompt of 128 tokens on Llama 2 7b.
>
>
> | **Response Length** | **Ours + SIGMA – Generation (LAN / WAN)** | **zkLLM – Generation** |
> |---------------------|--------------------------------------------|--------------------------|
> | 1   | 22.1 / 22.2       | 369.7   |
> | 2   | 44.4 / 44.6       | 726.5   |
> | 5   | 118.3 / 118.9     | 1798.1    |
> | 10  | 227.7 / 236.6     | 3889.3    |
> | 50  | 1189.0 / 1194.0   | 19550.8   |
> | 100 | 2260.0 / 2301.0   | 35860.1   |
>
> | **Response Length** | **Ours + SIGMA – Verify** | **zkLLM – Verify** |
> |---------------------|---------------------------|---------------------|
> | 1   | 0.17   | 1.24   |
> | 2   | 0.35   | 2.51   |
> | 5   | 0.91   | 6.32   |
> | 10  | 1.87   | 11.8   |
> | 50  | 3.73   | 59.2   |
> | 100 | 18.27  | 117.3  |
>
> As can be seen, our approach remains significantly faster for both generation and verification, even in the slower WAN setting, for long output responses, than the state of the art comparison, zkLLM.
>
> Our response is continued further in the next comment.

---

> ### Author Response · Authors · 2025-11-21
> **Response Part 2 to Reviewer Y7kc**
>
> W3: We actually don’t use the ‘honest-but-curious’ designation in our paper, as far as we are aware. However, it is a standard behavioral threat model in the literature; in simple terms, it models adversaries as being ‘honest’ insofar as they perform the requested operation faithfully, but that they are curious in that they will attempt to determine the content of the encrypted message they are operating on. This is the most frequently used behavioral assumption in privacy-preserving works, since such methods do not typically have an inbuilt means to enforce adversarial honesty (or detect adversarial dishonesty). That is precisely the shortcoming that we address; our protocols allow for the relaxation of the ‘honest’ assumption, which we believe is a much more plausible model of real-world adversaries.
>
> Regarding non-collusion, first, a distinction should be drawn between privacy schemes that rely on multiple parties (SMPC) and those which are single party (FHE, TEE). Collusion only pertains to the multiparty setting. It is a critical weakness of all SMPC schemes that they assume non-collusion (typically, at least one non-colluding party suffices) to maintain their privacy guarantees. For our protocols, their collusion resistance is inherited directly from the underlying privacy scheme. Therefore, for any FHE-based instantiation of our protocols, we are resistant to collusion (actually, there is no meaningful concept of collusion with a single party); and for any SMPC-based instantiation of our protocols, we are vulnerable to the setting where all parties collude. We emphasize however that our protocols are agnostic to the exact privacy scheme used, as long as they maintain computational indistinguishability of the inputs.
>
> Thank you for mentioning this point. We have now included the above in our current draft of the paper.
>
> Q3: This is a great question! The interaction of our protocols with TEEs is multifaceted and complex. TEEs come equipped with both privacy-preservation (via memory encryption of processes restricted to hardware enclaves) as well as a form of verifiability with an attestation mechanism. As we have outlined in Appendix A.1 of our paper, there are potential vulnerabilities in these attestations, and a wide body of literature covering them; and since the submission of our paper, there have been further significant attacks uncovered e.g. https://tee.fail/. These attacks have different properties w.r.t. whether they break confidentiality, verifiability, or both, and the exact extent to which they do either (e.g. some attacks allow for a limited degree of memory modification, but not sufficiently so to replace the entirety of a model’s loaded weights in a VM). Given the wide variety of attacks, and the very particular details around the adversarial behaviors they enable, we refrained from making broad statements regarding how our protocols would interact with TEE attack surfaces.
>
> However, a fundamental design component of TEEs is that the attestation service bears the trust assumption that the hardware vendor is the ultimate certificate authority for attestation, and will not maliciously sign fake enclave measurements. This assumption of attestational-integrity is relaxed by use of our protocols, enabling a lower-trust usage model of TEEs.
>
> Finally, we note that TEEs typically have significantly less overhead (on the order of 5-10% for LLM inference) than SMPC/FHE, and so are a very lightweight privacy-preserving inference method.
>
> We have now updated our paper to include the above further clarification of the interaction of our protocols with TEEs. Thank you again for asking this pertinent question.
>
> Our response is continued further in the next comment.

---

> ### Author Response · Authors · 2025-11-21
> **Response Part 3 to Reviewer Y7kc**
>
> Q4: Existing inference engines do not support privacy-preserving inference methods such as FHE or SMPC out of the box. However, there are several useful libraries, such as [Crypten](https://github.com/facebookresearch/CrypTen), that enable relatively straightforward inference with SMPC. The modifications required to support our protocols with Crypten are minimal, requiring less than 100 lines of core code. We open source our code at https://anonymous.4open.science/r/priveri/
>
> Q5: Yes, this is a great question. Our response above contains many of the key settings that our protocols are applicable to. To summarize, our protocols are beneficial in any setting where 1) privacy-preservation is already in use, but there is no accompanying verification technique; our protocols allow relaxation of the ‘honest-but-curious’ assumption without requiring complex and costly overheads by integrating ZK-proofs 2) TEEs are in use and there is a benefit from relaxing the hardware-vendor trust assumptions around attestations 3) a security level on the order of cryptographic guarantees is not wholly necessary, and statistical difficulty of the problem is sufficient. Private LLM inference is one such setting – the security requirement of these is typically less stringent than e.g. financial transactions, as the content of any single message does not have the same level of sensitivity to leakage. Therefore, our protocols are useful in any LLM serving setting where the user wishes to verify the model used by the inference provider. For example, there is currently no clear way for users to know if closed-source companies route their prompts to a less expensive model than the one they are paying for; our protocols would allow for the detection of this. Of course, this requires the closed-source company to cooperate in setting up the infrastructure for privacy-preservation in the first place, but for example, Google has signalled that they are moving in this direction, e.g. see this [private-computing report released last week](https://services.google.com/fh/files/misc/private_ai_compute_technical_brief.pdf). Similarly, our protocols are useful in any decentralized environment where untrusted third parties provide the inference compute, such as the [Gradient](https://gradient.network/) or [Akash](https://akash.network/) networks, for example.
>
> Thank you once again for your thorough review and your insightful comments. If you have any further questions or comments for any part of our responses, or if you think our submission can be further improved in any way, please let us know. We are happy to continue the discussion at any time until the end of the discussion period. We have made a significant effort to address each of your questions and weaknesses and would appreciate it if you would consider raising your score in light of our response. Thank you!

---

### Official Review · Reviewer_qDjq · 2025-11-02

**Soundness:** 1
**Presentation:** 2
**Contribution:** 1
**Rating:** 2
**Confidence:** 5

**Summary:**

he submission proposes several mechanisms to ensure the verifiability
of privacy-preserving LLM inferences. By "verifiability," the user who
supplies private input x can verify that the output y is legitimately
obtained by computing M(x). In the FHE scenario, the user may send a
ciphertext ct = Enc(x); the inference provider homomorphically evaluates
M on ct, and the user decrypts the resulting ciphertext to obtain y.
While one could naively achieve verifiability by using zero-knowledge
proofs, this solution tends to incur significant overhead.

The submission explores alternative and more lightweight methods:

- In Protocol 1, the user inserts randomly selected sentinel tokens into
the tokenized prompt at random positions. The user then observes the
output logit vector at all token positions and compares it against the
precomputed cached logits for model M.

- Protocol 2 is a variant of Protocol 1, which adds randomly sampled
noise to the token embeddings.

- In Protocol 3, the user appends a randomly generated key to the prompt
with explicit instructions asking to repeat the key in the response. The
user then checks that the response includes the same key.

**Strengths:**

The submission explores alternative solutions to verifiable LLM
inference. Since zero-knowledge proofs for verifiable LLM are
prohibitively expensive due to the underlying cryptographic operations,
proposing a new paradigm is a promising direction.

**Weaknesses:**

Unfortunately, the submission does not adequately justify the security
of the proposed methods.

- Protocol 3 does not provide a meaningful guarantee. Since the user
prompt always follows the same format, in the FHE setting, any malicious
inference provider can identify the location of the key in the text. By
applying suitable homomorphic operations, an adversary can single out
the ciphertext containing the key and append it to an arbitrary output
text. Thus, this methodology barely meets the requirement of verifiable
computation.

- The authors present some experiments in the SMPC setting, where a
prompt is secret-shared among multiple parties and at least one behaves
honestly. However, if the remaining parties are dishonest, one should
simply use SMPC with **active security** to achieve verifiability, as
the protocol can then tolerate any misbehavior. This means that as soon
as any dishonest party deviates from the protocol by trying to evaluate
secret shares of the private prompt on an incorrect circuit (i.e., wrong
model), the honest party can detect such cheating behavior.

- In general, the paper does not formally define what kind of
"verifiability" is guaranteed by the proposed solutions. I recommend
that the authors make the security notion mathematically precise. For
example, in the context of ZKP, verifiability can be formulated in a
game-based manner: for any input $x$ and model $M$, and any
(computationally bounded) adversary $A$ outputting a proof $\pi$ and
(encryption of) $y$, the probability $\Pr[M(x)\neq y \land V \text{
accepts } \pi]$ is negligible. SMPC with active security also formally
guarantees that either $M(x)=y$ is correctly computed (or otherwise the
protocol simply aborts if it does not have the guaranteed output
delivery property), even in the presence of malicious parties. Both
paradigms provide formal security proofs against _any_ adversarial
strategy. In contrast, the submission only checks security with respect
to _specific attack strategies_, which is generally not a sound
methodology for analyzing security.

**Questions:**

- Although the main goal of the paper is verifiability, the paper does
not provide any formal security notion as opposed to ZKP and SMPC. Could
the authors define a formal security definition similar to (knowledge)
soundness of ZKP or active security of SMPC?

- If one were to use SMPC to achieve privacy-preserving LLM inference,
wouldn't verifiability be trivially ensured by employing actively secure
SMPC? Note that actively secure SMPC does not necessarily require
zero-knowledge proofs.

---

> ### Author Response · Authors · 2025-11-21
>
> Thank you very much for your detailed review. We are glad that you find the new paradigm we introduce to be a promising direction.
>
> We address your other points below:
>
> W1: Thank you for raising this important point. We agree with your assessment that this is a valid attack on Protocol 3. As such, we have now removed Protocol 3 from our paper. We kindly request that you consider our paper’s contributions based on Protocol 1 and 2, which form the bulk of our submission and experimental work.
>
> W2 & Q2: This is a great question! Active security is indeed an inherent feature of SMPC schemes with at least one honest party. Our Protocols 1 and 2, however, are resistant even in the setting where *all* the parties are dishonest.
>
> To expand on this in detail, we enumerate the behavioral cases below:
> - If there is at least one honest party, we agree with your observation that using SMPC, one may rely on active security.
> - If all parties are dishonest _and_ colluding, then 1) active security fails 2) privacy can be broken by pooling the secret shares, so our Protocols 1 and 2 also are easy to bypass by observing the locations of the sentinel tokens.
> - If all parties are dishonest but a sufficient number are non-colluding (for many SMPC schemes, a single non-colluding party is sufficient), then 1) active security fails, as there is no honest party to raise the secret-share mismatch that arises 2) privacy is retained thanks to non-collusion, so our Protocols 1 and 2 retain their verification property.
>
> Therefore, our contribution improves on active security by relaxing the necessary assumptions in the SMPC setting. Further, in e.g. the FHE setting, there is no analogue to active security, and therefore our protocols address this shortcoming.
>
> We accept that we have not made the above point clearly in the paper as it stands. We have now updated our draft paper to make this point much more explicit. Thank you again for bringing this point to our attention, to allow us to revisit this and strengthen our work.
>
> W3 & Q1: Thank you for raising this point. Our protocol provides primarily _statistical_ guarantees, not cryptographic ones. As such, we cannot provide formal guarantees to the same level as ZKP or SMPC. This is also why our general approach is to consider possible attacks – and we test a wide range of them in our paper, including low-rank approximations, quantization, and fine-tuning. Although we accept this form of guarantee is weaker than a cryptographic one, we wish to point out an analogy to the adversarial robustness literature; in that field, the most performant algorithms (e.g. see [RobustBench](https://robustbench.github.io/)) generally do not have provable worst case guarantees, but are nevertheless the most useful and resistant to attacks in practice.
>
> Finally, we would like to draw to your attention that we have now open sourced an implementation of our Protocol 1 using the CrypTen library at https://anonymous.4open.science/r/priveri/. Further, we have extended our experiments in Section 7 to longer response lengths and different network conditions -- please see our response to Reviewer Y7kc for the details. We find that our protocols remain significantly faster for both generation and verification than zkLLM.
>
> Thank you once again for your thorough review and your insightful comments. If you have any further questions or comments for any part of our responses, or if you think our submission can be further improved in any way, please let us know. We are happy to continue the discussion at any time until the end of the discussion period. We have addressed your main concerns, and would appreciate it if you would consider raising your score in light of our response. Thank you!

---

### Author Response · Authors · 2025-12-01
**Meta-Rebuttal**

We thank all the reviewers for their constructive feedback on how to improve our submission. We note that all reviewers highlighted the novelty of our work. We have directly addressed all feedback from reviewers, including additional experiments, which we summarize below.

In response to reviewers **NATw** and **Y7kc**, we performed additional experiments of inference times for multiple token generation under our protocols. We found that even in slower network settings, our protocols reduce inference time by around 15x compared to the state-of-the-art zero-knowledge based proof-of-inference method, zkLLM. We found even larger speedups for verification runtimes.

Reviewers **Y7kc** and **qDjq** both commented on the guarantees that our protocols present. We have clarified that our protocols are not formally provable in terms of their guarantees, but instead, offer statistical guarantees, which we have tested against a wide variety of possible attacks.

In addition, we have now open-sourced our implementation of Protocol 1 using the CrypTen library at https://anonymous.4open.science/r/priveri/.

To conclude, our work is – to the best of our knowledge – the first to explore the possibility of utilizing privacy-preserving mechanisms for LLM inference in order to also obtain cheap verification of LLM inference. Our introduced protocols are robust to a wide variety of attacks, including being able to detect even a single gradient update to the model, and are much more performant than the current state of the art for verifiable LLM inference.

---

### Meta-Review · Area_Chair_ZcYR · 2025-12-08

**Summary:**

The paper leverages privacy-preserving mechanisms to achieve verifiable LLM inference at a lower cost than ZKPs. While the reviewers acknowledged the novelty of the direction and the authors provided extensive new experiments during the rebuttal to demonstrate performance benefits over zkLLM, significant concerns regarding soundness remain.

The primary rationale for the rejection is the lack of formal cryptographic security guarantees. The paper relies on statistical guarantees and ad-hoc defenses, which are insufficient for a contribution in the intersection of privacy and verification. Furthermore, the identification of a fatal flaw in Protocol 3 highlights the lack of theoretical rigor. Consequently, I recommend this paper for rejection.

**Reviewer Concerns:**

The rebuttal successfully addressed the empirical concerns raised by Reviewers NATw and Y7kc regarding performance. Additionally, the authors addressed the validity issues in Protocol 3 by removing it entirely after Reviewer qDjq identified fatal flaws.

However, critical theoretical concerns remain outstanding. The authors failed to provide the formal security definitions or cryptographic proofs requested by Reviewer qDjq, and their argument that statistical guarantees are sufficient is unconvincing in this context.

**Reviewer Scores:**

I anticipate that all reviewers would maintain their original scores. Reviewers qDjq and NATw would likely persist in their rejection, as  the fundamental lack of formal cryptographic guarantees remains unaddressed. Reviewer eGgA and Y7kc stated low confidence in their assessments of the security mechanisms.  Consequently, their scores are outweighed by the significant soundness concerns raised by Reviewers qDjq and NATw.

---

### Decision · Program_Chairs · 2026-01-26

Reject